# Practical Three-Factor Authentication Protocol Based on Elliptic Curve Cryptography for Industrial Internet of Things

**DOI:** 10.3390/s22197510

**Published:** 2022-10-03

**Authors:** Xingwen Zhao, Dexin Li, Hui Li

**Affiliations:** 1State Key Laboratory of Integrated Service Networks, Xidian University, Xi’an 710071, China; 2School of Cyber Engineering, Xidian University, Xi’an 710000, China

**Keywords:** industrial Internet of things, wireless sensor network, authentication and key agreement, elliptic curve cryptography, forward secrecy

## Abstract

Because the majority of information in the industrial Internet of things (IIoT) is transmitted over an open and insecure channel, it is indispensable to design practical and secure authentication and key agreement protocols. Considering the weak computational power of sensors, many scholars have designed lightweight authentication protocols that achieve limited security properties. Moreover, these existing protocols are mostly implemented in a single-gateway scenario, whereas the multigateway scenario is not considered. To deal with these problems, this paper presents a novel three-factor authentication and key agreement protocol based on elliptic curve cryptography for IIoT environments. Based on the elliptic curve Diffie–Hellman problem, we present a protocol achieving desirable forward and backward secrecy. The proposed protocol applies to single-gateway and is also extended to multigateway simultaneously. A formal security analysis is described to prove the security of the proposed scheme. Finally, the comparison results demonstrate that our protocol provides more security attributes at a relatively lower computational cost.

## 1. Introduction

The emerging industrial Internet of things (IIoT) is a typical application scenario for wireless sensor network (WSN), where the IIoT is dedicated to affording the capacity to construct innovative services and applications within the industrial automation scenario [1]. The IIoT emphasizes extremely low latency, high security, and the ability to handle massive quantities of data [2]. Therefore, efficient authentication and key agreement mechanisms should be designed for the IIoT infrastructure to ensure security and privacy. In this manner, only authorized principals can access the IIoT resource, and these legal entities can interact over the channel using the session key that they have negotiated.

Considering authentication protocols for sensors with a low computing power, the literature [3,4] sacrifices security to build lightweight protocols, resulting in these schemes being vulnerable to certain attacks. It is clearly found that schemes using only a hash function, exclusive OR (XOR), and symmetric cryptography are unable to achieve forward and backward secrecy. Ma et al. [5] claimed that the public key cryptography algorithm was indispensable to achieve forward secrecy. After that, public key cryptography technology was widely implemented in authentication protocols, where using elliptic curve cryptography (ECC) or bilinear pairings was able to help protocols achieve forward and backward secrecy.

Figure 1 illustrates that a representative IIoT architecture usually consists of three categories of entities: industrial IoT sensing devices, an industrial central, and an engineering expert [6], which, respectively, represent sensors, the gateway, and the user in WSNs. IIoT sensing devices are leveraged to monitor the status of objects and gather data, which is subsequently forwarded to a gateway via a wireless channel. A user is able to access the data collected by the gateway in real time. Sensors, in general, have low processing power, limited computational capabilities, and restricted energy and storage capacity, whereas gateways have a strong capacity for data processing [7].

### 1.1. Literature Review

Das [8] first presented a password and smart-card-based two-factor user authentication protocol for WSNs using merely the hash function in 2009. Since then, some drawbacks to this scheme have been discovered by scholars. The presented schemes [9,10,11] identified some vulnerabilities in Das’s scheme [8], and they suggested various countermeasures to overcome these flaws. In 2014, Turkanvoic et al. [12] proposed a novel user and mutual authentication scheme for WSNs using only a hash function and XOR. These lightweight schemes consumed relatively fewer resources but sacrificed security.

In order to achieve more security attributes, a public-key infrastructure was considered in some schemes. In 2011, Yeh et al. [13] performed a cryptanalysis of Das’s scheme [8], and they discovered that there was no mutual authentication and no protection against an insider attack or forgery attack. As a result, they first implemented ECC to build the authentication protocol to address the current existing weaknesses. Shi and Gong [14] proposed a new ECC-based authentication protocol for WSNs in 2013, which addressed the shortcomings of the scheme in [13] that lacked a key agreement and forward secrecy. In 2016, Chang and Le [15] stated briefly that the scheme from Turkanovic et al. [12] suffered from an impersonation attack, stolen smart card attack, stolen-verifier attack, and failed to ensure backward secrecy, and they proposed an advanced scheme that used ECC to overcome these flaws. In 2018, Li et al. [16] indicated that the protocol in [15] lacked a proper mutual authentication and had other functionality defects. They [16] presented a three-factor user authentication protocol for the IIoT that addressed the protocol’s [15] shortcomings by utilizing ECC and symmetric cryptography. A majority of protocols, however, are designed for a single-gateway scenario, ignoring how to implement them in a multigateway scenario.

In 2016, Aim and Biwas [17] solved some security flaws in the scheme from Turkanvoic et al. [12] and designed the first authentication protocols for a multigateway scenario. Later, Das et al. [18] indicated that there were no efficient online sensor node registration and password change phases in the literature [17], and they presented a new three-factor user authentication scheme applied to the multigateway WSN architecture using AES (Advanced Encryption Standard). In 2017, Wu et al. [19] demonstrated that the scheme in [17] suffered from tracking attacks due to the constant pseudo-identity and previously established session key that adversaries could calculate and presented a novel authentication scheme for multigateway WSNs. Srinivas et al. [20] showed that the protocol in [17] suffered from a stolen smart card attack, password guessing attack, and impersonation attack. They proposed an authentication scheme for multigateway WSNs that could withstand all the above-mentioned attacks. In 2018, Wang et al. [21] discovered that the scheme in [20] was still subject to offline password guessing attacks and node capture attacks and could not protect the user’s anonymity. Therefore, they described efficient countermeasures for these attacks. Since all the above-mentioned multigateway schemes use lightweight cryptographic primitives, it is impossible to achieve forward and backward secrecy. Accordingly, our scheme will solve this problem.

### 1.2. Network Model

Figure 2 demonstrates how the single-gateway model is implemented in our presented IIoT protocol. After the user logs in, they send the message to the home gateway node (HGWN). If the user can pass the authentication of the HGWN, the HGWN sends the message to the sensor. After the sensor authenticates, it computes the session key and sends a message to the HGWN. Finally, the HGWN sends a message to the user, who calculates the session key to communicate with the sensor. Through two rounds of complete information exchange, the user, HGWN, and sensor can realize mutual authentication.

Nevertheless, in traditional single-gateway WSNs, high-speed data streams are prone to conflict during data aggregation, because the distance between edge sensors and the gateway node is too far, which may cause an increased communication cost and reduced performance. In this case, multigateway protocols are required, and Figure 3 shows the model we used. This architecture is an extension of Figure 2. The user sends the authentication message to the HGWN. Following that, the HGWN checks the validity of the received message. In the event that this procedure is successful, the HGWN sends a message to the FGWN. The FGWN transmits a message to the HGWN after confirming the message’s availability. Then, the HGWN checks the received message and delivers a message to the user. Following steps 1–4, the mutual authentication is achieved between the user and the FWGN. After that, user sends a message to the FGWN for further authentication. After the verification is successful, the FGWN transmits a message to the sensor. Subsequently, the sensor computes the session key and delivers a message to the FGWN. Finally, the user figures out the session key used for subsequent communication after confirming the message that the FGWN sent to it.

### 1.3. Motivations and Contributions

1. Intractable elliptic curve Diffie–Hellman problem (ECDHP) is applied to our protocol to guarantee the security of the session key. We extend our scheme to multigateway WNSs while considering the limitations of single-gateway WSNs.

2. The random oracle model (ROM) [22] helps us get the formal proof of the presented scheme. The result indicates that the probability of an adversary who can break the proposed protocol is negligible.

3. Scyther, an automated security protocol verification tool [23], is used to simulate and analyze the proposed protocol. The result demonstrates that the scheme is correct and secure against many adversary models.

## 2. Preliminaries

### 2.1. Elliptic Curve Cryptography

ECC was initially proposed by Koblitz [24] and Miller [25] in the 1980s, and an introduction to the basic knowledge of ECC is described in the following. Given a large prime number *p* and a finite field $F_{p}$, let a set of elliptic curve points *E* over $F_{p}$ be defined by the equation: $E\left(F_{p}\right):y^{2}=x^{3}+a\cdot x+b\;\mathrm{mod}\;p$, where $a,b\in F_{p}$ and $\Delta =4a^{3}+27b^{2}\neq 0\;\mathrm{mod}\;p$. All points on $E(F_{p})$ and the point *O* at infinity come from an additive Abelian group *G* of order *q*, where *P* is the generator point of the group and n·P=P+P+…+P, where *n* is an integer and $n\in Z_{q}^{*}$. There are two corresponding mathematical problems in ECC defined as follows:The elliptic curve discrete logarithm problem (ECDLP): Figure 4 demonstrates points distributed over an elliptic curve $y^{2}=x^{3}-x+2$ in finite field $F_{97}$. Selecting two points *Q* and *P* in Figure 4, where $Q,P\in F_{97}$ satisfy Q=kP, where *k* is between 0 and 96 at random. Given *k* and *P*, it is easy to figure out *Q* by a scalar multiplication and addition rules. Nevertheless, given *Q* and *P*, it is difficult to calculate *k*.The elliptic curve Diffie–Hellman problem (ECDHP): It is scarcely possible to find abP when given $aP\in F_{p}$ and $bP\in F_{p}$ in polynomial time, where *a* and *b* are both between 0 and p−1 at random.

### 2.2. Threat Model

The proposed authentication and key agreement protocol was formally analyzed taking advantage of the Dolev–Yao threat model [26], which assumes that two communication principals interact over an insecure and open channel. The following are the properties of this model:The used one-way hash function is unbreakable.In a uniform protocol, an identical format is used by each entity that wishes to communicate.An adversary can eavesdrop, intercept, replay, and even modify all the transmitted messages over an open and insecure channel.

### 2.3. Fuzzy Extractor

Biometric features are adopted to improve security in many schemes. Due to the uniqueness of biometric features, they can be effectively applied to authentication. Compared with low-entropy passwords, biometric features also have the advantages of being difficult to forge and not being easy to lose.

The fuzzy extractor was used to process the original biometric fingerprint, which can eliminate subtle differences between biometric features extracted by the same user at different points in time. A fuzzy extractor comprises two phases as follows Ref. [27]:Probabilistic generation function Gen: The original biometric fingerprint $BIO_{i}$ is the input of Gen, and then the process outputs biometric identification key data and public parameter, namely $Gen\left(BIO_{i}\right)\to \left(\sigma_{i},\theta_{i}\right)$.Deterministic reproduction procedure Rep: Using the public parameter $\theta_{i}$ and the fingerprint $BIO_{i}$ reproduces key data $\sigma_{i}$, namely $Rep\left(BIO_{i},\theta_{i}\right)\to \sigma_{i}$.

## 3. The Proposed Scheme

In this section, the detailed process of the proposed scheme is demonstrated. The proposed scheme consists of the following phases: initialization phase, registration phase, user login phase, authentication and key agreement phase, and user password update phase.

### 3.1. Initialization Phase

All the parameters that are used in the proposed protocol are listed in Table 1. During the initialization phase, SA chooses an elliptic curve *E* over a prime finite field $F_{p}$, a point $P\in E(F_{p})$ and a subgroup *G* of $E(F_{p})$, where *G* is an additive cyclic group of order *q*. Then, the HGWN generates its private key and public key $\left\{k_{h},K_{h}\right\}$, where $k_{h}\in Z_{q}^{*}$ and $K_{h}=k_{h}P$. Consistent with the above procedure, the FGWN chooses its private key and public key $\left\{k_{f},K_{f}\right\}$, where $k_{f}\in Z_{q}^{*}$ and $K_{f}=k_{f}P$. Finally, the hash function $h\left(\cdot \right):{\left\{0,1\right\}}^{*}\to {\left\{0,1\right\}}^{l}$ is chosen to be used in the scheme, where *l* is the length of the output length of the hash function.

### 3.2. Registration Phase

The registration phase is divided into a user registration phase and a sensor registration phase. All the messages in this phase are transmitted via a secure channel.

#### 3.2.1. User Registration Phase

The procedure is also shown in Figure 5.

**Step 1**: $U_{i}$ selects their identity $ID_{i}$ and password $PW_{i}$, and inputs biometric information $BIO_{i}$. The fuzzy extractor is used to compute biometric key data $\sigma_{i}$ and public parameter $\theta_{i}$, namely $Gen\left(BIO_{i}\right)\to \left(\sigma_{i},\theta_{i}\right)$. $SC_{i}$ stores the public parameter $\theta_{i}$ in its memory. Then, $U_{i}$ figures out $HID_{i}=h(ID_{i}\left|\right|\sigma_{i})$ and $HPW_{i}=h(PW_{i}\left|\right|\sigma_{i})$, and sends $\left\{HID_{i},HPW_{i}\right\}$ to the nearest HGWN via a secure channel.

**Step 2**: Upon receiving $\left\{HID_{i},HPW_{i}\right\}$ from $U_{i}$, the HGWN generates a random number $r_{h}$ and calculates $A_{i}=h(HID_{i}\left|\right|k_{h}\left|\right|r_{h})⊕HID_{i}$, $B_{i}=h(HID_{i}||HPW_{i}\left|\right|r_{h})$, and $C_{i}=HID_{i}⊕r_{h}$. The HGWN stores $\left\{HID_{i},r_{h}\right\}$ in its memory. Then, the HGWN sends $\left\{A_{i},B_{i},C_{i}\right\}$ to $U_{i}$ via a secure channel.

**Step 3**: Upon getting $\left\{A_{i},B_{i},C_{i}\right\}$ from HGWN, $U_{i}$ stores $\left\{A_{i},B_{i},C_{i},\theta_{i}\right\}$ into its own $SC_{i}$.

#### 3.2.2. Sensor Registration Phase

Sensor registration process is shown in Figure 6. SA assigns a unique identity to each sensor node. $SN_{j}$ sends its own identity $SID_{j}$ to the nearest HGWN via a secure channel for registration. Then, the HGWN calculates $A_{gs}=h(SID_{j}\left|\right|k_{h})$ and stores $\left\{SID_{j},A_{gs}\right\}$ in its memory. After that, the HGWN sends $A_{gs}$ to $SN_{j}$ via a secure channel. After receiving $A_{gs}$ from the HGWN, $SN_{j}$ stores $\left\{SID_{j},A_{gs}\right\}$ in its own memory.

### 3.3. User Login Phase

$U_{i}$ inserts their smart card $SC_{i}$ to a terminal, and inputs identity $ID_{i}$, password $PW_{i}$ and biometric information $BIO_{i}$. Then, the terminal reproduces the biometric key data $\sigma_{i}$ through the fuzzy extractor, namely $Rep\left(BIO_{i},\theta_{i}\right)\to \sigma_{i}$. The terminal computes $HID_{i}=h(ID_{i}\left|\right|\sigma_{i})$, $HPW_{i}=h(PW_{i}\left|\right|\sigma_{i})$, $r_{h}^{\prime }=HID_{i}⊕C_{i}$ and $B_{i}^{\prime }=h(HID_{i}||HPW_{i}\left|\right|r_{h}^{\prime })$. Subsequently, the terminal checks whether $B_{i}^{\prime }\overset{?}{=}B_{i}$. If the equation is not held, at least one parameter is incorrect, which leads to the login request being refused by the terminal and no subsequent authentication process being performed. Otherwise, $U_{i}$’s login is successful, and the terminal generates a random number $a\in Z_{q}^{*}$, and a timestamp $T_{1}$. At last, the terminal computes $A_{h}=A_{i}⊕HID_{i}$, $D_{1}=aP$, $D_{2}=aK_{h}$, $M_{1}=HID{}_{i}⊕h\left(D_{2}\right)$, $M_{2}=SID_{j}⊕h\left(D_{2}\right)⊕A_{h}$, and $M_{3}=h(HID_{i}\left|\right|A_{h}\left|\right|D_{2}\left|\right|M_{1}\left|\right|M_{2}\left|\right|T_{1})$. This process is demonstrated in Figure 7.

### 3.4. Authentication and Key Agreement Phase

In this section, two cases are considered: authentication and key agreement in a home region and a foreign region, respectively.

#### 3.4.1. Authentication and Key Agreement in the HGWN

When a user and the sensor that they want to access are in the same region controlled by the same HGWN, as illustrated in Figure 8, each entity will execute the following steps.

**Step 1**: $U_{i}$ sends the login request message $\left\{M_{1},M_{2},M_{3},D_{1},T_{1}\right\}$ to the HGWN.

**Step 2**: After receiving $\left\{M_{1},M_{2},M_{3},D_{1},T_{1}\right\}$ from $U_{i}$, the HGWN checks whether $|T_{1}^{\prime }-T_{1}|<\Delta T$ is satisfied, where $T_{1}^{\prime }$ is the current timestamp the HGWN acquired and ΔT is the acceptable maximum transmission delay. If the inequality is not true, namely $T_{1}$ is not fresh, the HGWN aborts the current session. Otherwise, the HGWN computes $D_{2}^{\prime }=k_{h}D_{1}$ and $HID_{i}^{\prime }=M_{1}⊕h\left(D_{2}^{\prime }\right)$ to find $r_{h}$ stored in its own memory. Subsequently, the HGWN calculates $A_{h}^{\prime }=h(HID_{i}^{\prime }\left|\right|k_{h}\left|\right|r_{h})$, $SID_{j}^{\prime }=M_{2}⊕h\left(D_{2}^{\prime }\right)⊕A_{h}^{\prime }$, and $M_{3}^{\prime }=h(HID_{i}^{\prime }\left|\right|A_{h}^{\prime }\left|\right|D_{2}^{\prime }\left|\right|M_{1}\left|\right|M_{2}\left|\right|T_{1})$, and checks whether $M_{3}^{\prime }\overset{?}{=}M_{3}$. The current session is aborted if $M_{3}^{\prime }\neq M_{3}$. Otherwise, the HGWN seeks $A_{gs}$ from its own memory through $SID_{j}$, generates a random number $r_{hg}$, a timestamp $T_{2}$, and calculates $M_{4}=r_{hg}⊕h(A_{gs}\left|\right|T_{2})$, $M_{5}=h(SID_{j}\left|\right|r_{hg}\left|\right|A_{gs}\left|\right|D_{1}\left|\right|T_{2})$. Finally, the HGWN sends $\left\{M_{4},M_{5},D_{1},T_{2}\right\}$ to $SN_{j}$.

**Step 3**: When ${SN}_{j}$ receives $\left\{M_{4},M_{5},D_{1},T_{2}\right\}$ from the HGWN, ${SN}_{j}$ obtains the current timestamp $T_{2}^{\prime }$ and verifies whether $|T_{2}^{\prime }-T_{2}|<\Delta T$. If the inequality is not held, then ${SN}_{j}$ terminates the current session. Otherwise, ${SN}_{j}$ figures out $r_{hg}^{\prime }=h(A_{gs}\left|\right|T_{2})⊕M_{4}$, $M_{5}^{\prime }=h(SID_{j}\left|\right|r_{hg}^{\prime }\left|\right|A_{gs}\left|\right|D_{1}\left|\right|T_{2})$, and examines whether $M_{5}^{\prime }\overset{?}{=}M_{5}$. The current session is terminated if $M_{5}^{\prime }\neq M_{5}$. Otherwise, ${SN}_{j}$ generates a random number $b\in Z_{q}^{*}$, a timestamp $T_{3}$, and figures out $D_{3}=bP$, $D_{4}=bK_{h}$, $SK=h(D_{1}||D_{3}||bD_{1})$, $M_{6}=h(SID_{j}\left|\right|r_{hg}\left|\right|A_{gs}\left|\right|D_{4}\left|\right|T_{3})$, and $M_{7}=h(SK||D_{1}\left|\right|D_{3})$. Lastly, ${SN}_{j}$ transmits $\left\{M_{6},M_{7},D_{3},T_{3}\right\}$ to the HGWN.

**Step 4**: After getting $\left\{M_{6},M_{7},D_{3},T_{3}\right\}$ from ${SN}_{j}$, the HGWN acquires the current timestamp $T_{3}^{\prime }$ and verifies whether $|T_{3}^{\prime }-T_{3}|<\Delta T$. If the verification fails, the HGWN aborts the current session. Otherwise, the HGWN calculates $D_{4}^{\prime }=k_{h}D_{3}$, $M_{6}^{\prime }=$$h(SID_{j}||r_{hg}||A_{gs}||D_{4}^{\prime }\left|\right|T_{3})$, and checks whether $M_{6}^{\prime }\overset{?}{=}M_{6}$. If $M_{6}^{\prime }\neq M_{6}$, the HGWN aborts the current session. Otherwise, the HGWN generates a timestamp $T_{4}$, calculates $M_{8}=h(HID_{i}\left|\right|A_{h}\left|\right|D_{1}\left|\right|D_{3}\left|\right|M_{7}\left|\right|T_{4})$, and dispatches $\left\{M_{7},M_{8},D_{3},T_{4}\right\}$ to $U_{i}$.

**Step 5**: Upon receiving $\left\{M_{7},M_{8},D_{3},T_{4}\right\}$ from the HGWN, $U_{i}$ obtains the current timestamp $T_{4}^{\prime }$ and checks whether $|T_{4}^{\prime }-T_{4}|<\Delta T$. If the verification fails, the current session is rejected by $U_{i}$. Otherwise, $U_{i}$ computes $M_{8}^{\prime }=h(HID_{i}\left|\right|A_{h}\left|\right|D_{1}\left|\right|D_{3}\left|\right|M_{7}\left|\right|T_{4})$ and checks whether $M_{8}^{\prime }\overset{?}{=}M_{8}$. If $M_{8}^{\prime }\neq M_{8}$, $U_{i}$ aborts the current session. Otherwise, $U_{i}$ computes $SK^{\prime }=h(D_{1}\left|\right|D_{3}||aD_{3})$, $M_{7}^{\prime }=h(SK^{\prime }\left|\right|D_{1}\left|\right|D_{3})$, and verifies whether $M_{7}^{\prime }\overset{?}{=}M_{7}$. If not, $U_{i}$ declines to establish a session key with ${SN}_{j}$. Otherwise, $U_{i}$ and ${SN}_{j}$ share an identical session key, and the authentication process is successfully completed.

#### 3.4.2. Authentication and Key Agreement in the FGWN

When a user requires access to a sensor that is in a foreign region and registered in a FGWN, this phase can be completed with the assistance of the HGWN and the FGWN, as illustrated in Figure 9 and Figure 10.

**Step 1**: $U_{i}$ computes the login request message $\left\{M_{1},M_{2},M_{3},D_{1},T_{1}\right\}$ as in the User Login Phase Section and sends them to the HGWN.

**Step 2**: After receiving $\left\{M_{1},M_{2},M_{3},D_{1},T_{1}\right\}$ from $U_{i}$, the HGWN obtains the current timestamp $T_{1}^{\prime }$ and verifies $T_{1}$’s validity, namely $|T_{1}^{\prime }-T_{1}|<\Delta T$. If the verification fails, the HGWN aborts. Otherwise, the HGWN calculates $D_{2}^{\prime }=k_{h}D_{1}$, $HID_{i}^{\prime }=M_{1}⊕h\left(D_{2}^{\prime }\right)$, $A_{h}^{\prime }=h(HID_{i}^{\prime }\left|\right|k_{h}\left|\right|r_{h})$, $SID_{j}^{\prime }=M_{2}⊕A_{h}^{\prime }⊕h\left(D_{2}^{\prime }\right)$, and $M_{3}^{\prime }=h(HID_{i}^{\prime }\left|\right|A_{h}^{\prime }\left|\right|D_{2}^{\prime }\left|\right|M_{1}\left|\right|M_{2}\left|\right|T_{1})$. Subsequently, the HGWN checks whether $M_{3}^{\prime }\overset{?}{=}M_{3}$. The current session is aborted if $M_{3}^{\prime }\neq M_{3}$. Next, if ${SID}_{j}$ is not in the HGWN’s database, the HGWN broadcasts the target sensor’s identity ${SID}_{j}$ to the rest of the gateway nodes. If any FGWN finds ${SID}_{j}$ in its database, it will react to the HGWN and broadcasts its own public key $K_{f}$ in WSNs. Subsequently, the HGWN generates a random number $b\in Z_{q}^{*}$, timestamp $T_{2}$, and computes $D_{3}=bP$, $D_{4}=bK_{f}$, $\left(b+k_{h}\right)K_{f}$, and $M_{4}=h(SID_{j}\left|\right|D_{3}\left|\right|\left(b+k_{h}\right)K_{f}\left|\right|T_{2})$. Finally, the HGWN dispatches $\left\{M_{4},D_{3},T_{2}\right\}$ to the corresponding FGWN.

**Step 3**: Upon receiving $\left\{M_{4},D_{3},T_{2}\right\}$ from the HGWN, the corresponding FGWN obtains the current timestamp $T_{2}^{\prime }$ and verifies whether $|T_{2}^{\prime }-T_{2}|<\Delta T$. If not, the FGWN terminates the current session. Otherwise, the FGWN computes $D_{4}^{\prime }=k_{f}D_{3}$, $D_{4}^{\prime }+k_{f}K_{h}$, and $M_{4}^{\prime }=h(SID_{j}\left|\right|D_{3}\left|\right|D_{4}^{\prime }+k_{f}K_{h}\left|\right|T_{2})$, and examines whether $M_{4}^{\prime }\overset{?}{=}M_{4}$. the FGWN terminates the current session if $M_{4}^{\prime }\neq M_{4}$. Otherwise, the FGWN generates random numbers $c\in Z_{q}^{*}$, $r_{f}$, a timestamp $T_{3}$, and calculates $D_{5}=cP$, $D_{6}=cK_{h}$, $\left(c+k_{f}\right)K_{h}$, $A_{f}=h(HID_{i}\left|\right|k_{f}\left|\right|r_{f})$, $M_{5}=A_{f}⊕h\left(D_{6}\right)$, and $M_{6}=h(SID_{j}\left|\right|A_{f}\left|\right|\left(c+k_{f}\right)K_{h}\left|\right|M_{5}\left|\right|T_{3})$. Then, the FWGN transmits $\left\{M_{5},M_{6},D_{5},T_{3}\right\}$ to the HGWN.

**Step 4**: Upon getting $\left\{M_{5},M_{6},D_{5},T_{3}\right\}$ from the FGWN, the HGWN acquires the current timestamp $T_{3}^{\prime }$ and verifies whether $|T_{3}^{\prime }-T_{3}|<\Delta T$. If the verification fails, the HGWN rejects the current session. Otherwise, the HGWN figures out $D_{6}^{\prime }=k_{h}D_{5}$, $D_{6}^{\prime }+k_{h}K_{f}$, $A_{f}^{\prime }=M_{5}⊕h\left(D_{6}^{\prime }\right)$, and $M_{6}^{\prime }=h(SID_{j}\left|\right|A_{f}^{\prime }\left|\right|D_{6}^{\prime }+k_{h}K_{f}\left|\right|M_{5}\left|\right|T_{3})$, and checks whether $M_{6}^{\prime }\overset{?}{=}M_{6}$. If $M_{6}^{\prime }\neq M_{6}$, the HGWN rejects the current session. Otherwise, the HGWN generates a timestamp $T_{4}$, calculates $M_{7}=A_{f}⊕A_{h}$, $M_{8}=h(HID_{i}||SID_{j}\left|\right|A_{h}\left|\right|A_{f}\left|\right|M_{7}\left|\right|T_{4})$, and dispatches $\left\{M_{7},M_{8},T_{4}\right\}$ to $U_{i}$.

**Step 5**: After receiving $\left\{M_{7},M_{8},T_{4}\right\}$ from the HWGN, $U_{i}$ gets the current timestamp $T_{4}^{\prime }$ and checks whether $|T_{4}^{\prime }-T_{4}|<\Delta T$. If not, the current session is rejected by $U_{i}$. Otherwise, $U_{i}$ computes $A_{f}^{\prime }=M_{7}⊕A_{h}$, $M_{8}^{\prime }=h(HID_{i}||SID_{j}\left|\right|A_{h}\left|\right|A_{f}^{\prime }\left|\right|M_{7}\left|\right|T_{4})$ and checks whether $M_{8}^{\prime }\overset{?}{=}M_{8}$. If $M_{8}^{\prime }\neq M_{8}$, $U_{i}$ rejects the current session. Otherwise, $U_{i}$ generates a timestamp $T_{5}$ and computes $D_{2f}=aK_{f}$, $M_{9}=HID_{i}⊕h\left(D_{2f}\right)$, $M_{10}=h(HID_{i}\left|\right|A_{f}\left|\right|D_{2f}\left|\right|M_{9}\left|\right|T_{5})$, and delivers $\left\{M_{9},M_{10},T_{5}\right\}$ to the FGWN.

**Step 6**: After receiving $\left\{M_{9},M_{10},T_{5}\right\}$ from $U_{i}$, the FGWN obtains the current timestamp $T_{5}^{\prime }$ and checks whether $|T_{5}^{\prime }-T_{5}|<\Delta T$ is satisfied. If failed, the FGWN aborts the current session. Otherwise, the FGWN computes $D_{2f}^{\prime }=k_{f}D_{1}$, $HID_{i}^{\prime }=M_{9}⊕h\left(D_{2f}^{\prime }\right)$, $M_{10}^{\prime }=h(HID_{i}^{\prime }\left|\right|A_{f}\left|\right|D_{2f}^{\prime }\left|\right|M_{9}\left|\right|T_{5})$, and checks whether $M_{10}^{\prime }\overset{?}{=}M_{10}$. The current session is aborted if $M_{10}^{\prime }\neq M_{10}$. Otherwise, FGWN generates a random number $r_{fg}$, a timestamp $T_{6}$, and calculates $M_{11}=r_{fg}⊕h(A_{fs}\left|\right|T_{6})$, $M_{12}=h(SID_{j}\left|\right|r_{fg}\left|\right|A_{fs}\left|\right|D_{1}\left|\right|T_{6})$. Finally, FGWN sends $\left\{M_{11},M_{12},T_{6}\right\}$ to $SN_{j}$.

**Step 7**: When $SN_{j}$ receives $\left\{M_{11},M_{12},T_{6}\right\}$ from the FGWN, $SN_{j}$ obtains the current timestamp $T_{6}^{\prime }$ and verifies whether $|T_{6}^{\prime }-T_{6}|<\Delta T$. If not, $SN_{j}$ aborts the current session. Otherwise, $SN_{j}$ computes $r_{fg}^{\prime }=h(A_{fs}\left|\right|T_{6})⊕M_{11}$, $M_{12}^{\prime }=h(SID_{j}\left|\right|r_{fg}^{\prime }\left|\right|A_{fs}\left|\right|D_{1}\left|\right|T_{6})$, and examines whether $M_{12}^{\prime }\overset{?}{=}M_{12}$. The current session is aborted if $M_{12}^{\prime }\neq M_{12}$. Otherwise, $SN_{j}$ generates a random number $d\in Z_{q}^{*}$, a timestamp $T_{7}$, and figures out $D_{7}=dP$, $D_{8}=dK_{f}$, $SK=h(D_{7}||dD_{1})$, $M_{13}=h(SID_{j}\left|\right|r_{fg}\left|\right|A_{fs}\left|\right|D_{8}\left|\right|T_{7})$, and $M_{14}=h(SK||D_{7})$. After that, $SN_{j}$ transmits $\left\{M_{13},M_{14},D_{7},T_{7}\right\}$ to the FGWN.

**Step 8**: After getting $\left\{M_{13},M_{14},D_{7},T_{7}\right\}$ from $SN_{j}$, the FGWN acquires the current timestamp $T_{7}^{\prime }$ and verifies whether $|T_{7}^{\prime }-T_{7}|<\Delta T$. If the verification fails, the FGWN aborts the current session. Otherwise, the FGWN computes $D_{8}^{\prime }=k_{f}D_{7}$, $M_{13}^{\prime }=$$h(SID_{j}||r_{fg}||A_{fs}||D_{8}^{\prime }\left|\right|T_{7})$, and checks whether $M_{13}^{\prime }\overset{?}{=}M_{13}$. If $M_{13}^{\prime }\neq M_{13}$, the FGWN aborts the current session. Otherwise, the FGWN generates a timestamp $T_{8}$, calculates $M_{15}=h(HID_{i}\left|\right|A_{f}\left|\right|D_{1}\left|\right|D_{7}\left|\right|M_{14}\left|\right|T_{8})$, and dispatches $\left\{M_{14},M_{15},D_{7},T_{8}\right\}$ to $U_{i}$.

**Step 9**: After receiving $\left\{M_{14},M_{15},D_{7},T_{8}\right\}$ from the FGWN, $U_{i}$ obtains the current timestamp $T_{8}^{\prime }$ and checks whether $|T_{8}^{\prime }-T_{8}|<\Delta T$. If not, $U_{i}$ rejects the current session. Otherwise, $U_{i}$ computes $M_{15}^{\prime }=h(HID_{i}\left|\right|A_{f}\left|\right|D_{1}\left|\right|D_{7}\left|\right|M_{14}\left|\right|T_{8})$ and checks whether $M_{15}^{\prime }\overset{?}{=}M_{15}$. If $M_{15}^{\prime }\neq M_{15}$, $U_{i}$ aborts the current session. Otherwise, $U_{i}$ figures out $SK^{\prime }=h(D_{7}||aD_{7})$, $M_{14}^{\prime }=h(SK^{\prime }\left|\right|D_{7})$, and verifies whether $M_{14}^{\prime }\overset{?}{=}M_{14}$. If the verification fails, $U_{i}$ declines to establish a session key with $SN_{j}$. Otherwise, $U_{i}$ and $SN_{j}$ share an identical session key, and the authentication process is successfully completed.

### 3.5. User Password Update Phase

$U_{i}$ inserts their smart card $SC_{i}$ into the terminal, and enters identity $ID_{i}$, password $PW_{i}$, and biometric information $BIO_{i}$. Then, the terminal reproduces the biometric key data $Rep\left(BIO_{i},\theta_{i}\right)\to \sigma_{i}$ and reads secret parameter $C_{i}=HID_{i}⊕r_{h}$ in $SC_{i}$ to calculate $HID_{i}=h(ID_{i}\left|\right|\sigma_{i})$, $HPW_{i}=h(PW_{i}\left|\right|\sigma_{i})$, and $r_{h}^{\prime }=HID_{i}⊕C_{i}$. Next, the terminal checks $B_{i}\overset{?}{=}h(HID_{i}||HPW_{i}\left|\right|r_{h}^{\prime })$. If the equation is not held, this update request is rejected. Otherwise, this request is acknowledged, and the subsequent phase is performed. In the update phase, $U_{i}$ enters a new password $PW_{i}^{new}$. Subsequently, the terminal computes $HPW_{i}^{new}=h(PW_{i}^{new}\left|\right|\sigma_{i})$ and updates $B_{i}^{new}=h(HID_{i}||HPW_{i}^{new}\left|\right|r_{h}^{\prime })$ in $SC_{i}$.

## 4. Security Analysis

### 4.1. Formal Security Proof

The security of our protocol is proved under the ROM.

#### 4.1.1. Formal Security Model

The security of the presented protocol dependent on the CK model [28].

Participants: In this model, the adversary A controls the communication between all participants. For the single-gateway scenario, there are three types of participants in this protocol *P*: the user *U*, the gateway HGWN, and the sensor SN. Each principal has a large number of instances, which are usually treated as the actions of specific protocols run by each principal. $U^{i}$, $HGWN^{k}$, and $SN^{j}$ represent the *i*th instance of *U*, *k*th instance of HGWN, and *j*th instance of SN in *P* separately. Moreover, *I* denotes any other instance.

Queries: The interaction between A and the protocol principals occurs merely through oracle queries, which simulate A’s capabilities to break *P* in a real attack. A is allowed to execute the following queries.

$Execute(U^{i},HGWN^{k},SN^{j})$: A uses this query to simulate a passive attack, and they can obtain the entire transcript as a result of the conversation among *U*, HGWN, and SN.

$Send(I^{i},m)$: It models an active attack of A, who forges a message *m* and sends it to instance $I^{i}$. Subsequently, $I^{i}$ returns the processing outcomes of the message *m* to A according to *P*. If the message *m* is invalid, the query is ignored.

$SKReveal(I^{i})$: This query simulates that A can obtain session key SK of any completed session.

$SSReveal(I^{i})$: This query can be asked of an incomplete session and receives the internal state in return.

$Corrupt(I^{i})$: This query can help A obtain the private key of $I^{i}$, which is usually used to simulate the forward secrecy of protocols. A can obtain the private key of *U*, HGWN, and SN.

$Test(I^{i})$: A asks this query to a fresh instance. Then, A can continue to ask other queries, as long as the tested session remains fresh. In other words, if $I^{i}$ has been asked $SSReveal(I^{i})$, $SKReveal(I^{i})$, or $Corrupt(I^{i})$, both $I^{i}$ and its partner cannot be asked by a Test query.

$Test(I^{i})$ query is used to evaluate the semantic security of a session key. Only one test query is allowed to be executed during the whole game. To answer the test query, we imagine a challenger who flips a coin to define a bit *b*. If there is no session key established for instance $I^{i}$, then ⊥ is returned. If the query has already been asked, then it outputs the same answer as above. Otherwise, if b=1, $I^{i}$ returns the real session key. If b=0, $I^{i}$ returns an entirely random string of the same length as the session key. The final output of $Test(I^{i})$ is a bit $b^{\prime }$, which is the guessing value of *b*. The adversary wins this game if and only if $b^{\prime }=b$.

#### 4.1.2. Security Proof

Suppose A is the adversary who can break protocol *P* in polynomial time. $q_{hash}$ and $q_{send}$ refer to the number of hash query oracles and send query oracles, respectively. $Adv_{P}^{ECDHP}\left(t\right)$ represents the advantage of an adversary who can resolve the intractable ECDHP in polynomial time. Now, the advantage of A that breaks the semantic security of our authentication and key agreement (AKA) protocol is defined:(1)$$\begin{array}{c} Adv_{P}^{AKA}\left(A\right)\leq \frac{q_{hash}^{2}}{2^{l}}+\frac{q_{send}}{2^{l-1}}+2Adv_{P}^{ECDHP}\left(t\right) \end{array}$$

**Proof.** Game *i* (*i* = 0, 1, 2, 3, 4) is used to perform the whole procedure of *P*. The event $WG_{i}$ signifies that A guesses the bit *b* correctly to win the game. □

**Game 0**: In the random oracle model, the real attack on *P* is modeled, and the following formula can be obtained:(2)$$\begin{array}{c} Adv_{P}^{AKA}\left(A\right)=\left|2\mathrm{Pr}\left[WG_{0}\right]-1\right| \end{array}$$

**Game 1**: A carries out Execute queries to model an eavesdropping attack. Even if we take Execute queries into consideration, the probability of an adversary who can win the game has not increased.
(3)$$\begin{array}{c} Pr\left[WG_{1}\right]=Pr\left[WG_{0}\right] \end{array}$$

**Game 2**: Hash oracles are added to the foundation of Game 1 by Game 2. This game models the active attack, and A attempts to trick a legitimate principal into accepting the modified message. When the collision happens between the constructed information and the real authentication information, A gets the secret information and wins the game. According to the birthday paradox, the maximum probability of the hash oracle collision is $\frac{q_{hash}^{2}}{2^{l+1}}$, and we have:(4)$$\begin{array}{c} |Pr\left[WG_{2}\right]-Pr\left[WG_{1}\right]|\leq \frac{q_{hash}^{2}}{2^{l+1}} \end{array}$$

**Game 3**: Send queries are added. This game models the active attack, and A attempts to trick a legitimate principal into accepting the modified message. Therefore, we have:(5)$$\begin{array}{c} |Pr\left[WG_{3}\right]-Pr\left[WG_{2}\right]|\leq \frac{q_{send}}{2^{l}} \end{array}$$

**Game 4**: In this game, A asks Execute queries eavesdropping on all exchanged messages $\left\{M_{1},M_{2},M_{3},D_{1},T_{1}\right\}$, $\left\{M_{4},M_{5},D_{1},T_{2}\right\}$, $\left\{M_{6},M_{7},D_{3},T_{3}\right\}$, and $\left\{M_{7},M_{8},D_{3},T_{4}\right\}$. A executes $Corrupt(I^{i})$ to obtain the private key of this entity, where *I* is equal to *U*, HGWN, and SN successively, and thus A can obtain all the private keys. $SKReveal(I^{i})$ can be executed in this game. It will answer an SK if the target instance has formed an SK. A executes $SSReveal(I^{i})$ to get the internal state of an incomplete session. In order to compute the session key, A has to resolve the intractable ECDHP to get *a* or *b* from $D_{1}=aP$ or $D_{3}=bP$. Let $Adv_{P}^{ECDHP}\left(t\right)$ be the advantage of A, who can resolve the ECDHP in polynomial time. As a result, we get:(6)$$\begin{array}{c} |Pr\left[WG_{4}\right]-Pr\left[WG_{3}\right]|\leq Adv_{P}^{ECDHP}\left(t\right) \end{array}$$

At the end of Game 4, all the queries are simulated, so what A can do is to guess the bit *b* to win the game after performing Test query. Now, we have the following:(7)$$\begin{array}{c} Pr\left[WG_{4}\right]=\frac{1}{2} \end{array}$$

According to Equations (Equation 2)–(Equation 7), we can obtain Equation (Equation 1). It indicates that the adversary has negligible advantage in winning the game. Therefore, our protocol is secure under the random oracle model.

### 4.2. Formal Verification Using Scyther

Scyther is a tool for the formal analysis of security protocols under the perfect cryptography assumption, in which it is assumed that all cryptographic functions are perfect. In this section, we formally analyze the security of the proposed protocol based on Scyther in the HGWN and FGWN. The results in Figure 11 and Figure 12 illustrate that the scheme is correct and secure against many adversary models under the Scyther security checks.

### 4.3. Informal Security Analysis

#### 4.3.1. Mutual Authentication

In the home region, the HGWN authenticates $U_{i}$ by relying on $M_{3}=h(HID_{i}\left|\right|A_{h}\left|\right|D_{2}\left|\right|$$M_{1}$
$\left|\right|M_{2}\left|\right|T_{1})$, where $D_{2}$ is possessed by $U_{i}$ and can be recovered by the HGWN from $D_{1}$ and its private key $k_{h}$. $U_{i}$ authenticates the HGWN using $A_{h}$ contained in $M_{8}=h(HID_{i}\left|\right|A_{h}\left|\right|D_{1}\left|\right|D_{3}\left|\right|M_{7}$
$\left|\right|T_{4})$, which can only be calculated by $U_{i}$ and HGWN. Any other principals cannot obtain $A_{h}$. The HGWN verifies $SN_{j}$ dependent on $M_{6}=h(SID_{j}\left|\right|r_{hg}\left|\right|A_{gs}\left|\right|$$D_{4}\left|\right|T_{3})$, where $D_{4}$ is possessed by $SN_{j}$ and can be recovered by the HGWN from $D_{3}$ and $k_{h}$. $SN_{j}$ verifies the HGWN using $A_{gs}$ contained in $M_{5}=h(SID_{j}\left|\right|r_{hg}\left|\right|A_{gs}\left|\right|D_{1}\left|\right|T_{2})$, which can be calculated by the HGWN and stored in $SN_{j}$’s memory. $U_{i}$ can verify the legitimacy of SK using $M_{7}$.

In the foreign region, there is a similar process as above. The HGWN authenticates $U_{i}$ by relying on the secret parameter $D_{2}$ only shared by both parties. $U_{i}$ authenticates the HGWN using $A_{h}$ contained in $M_{8}=h(HID_{i}||SID_{j}\left|\right|A_{h}\left|\right|A_{f}\left|\right|M_{7}\left|\right|T_{4})$. The FGWN and HGWN implement mutual authentication using $\left(b+k_{h}\right)K_{f}$ and $\left(c+k_{f}\right)K_{h}$, respectively, which are both the secret parameters and can only be computed by themselves and verified by the other party. The FGWN authenticates $U_{i}$ dependent on $M_{10}=h(HID_{i}\left|\right|A_{f}\left|\right|D_{2f}\left|\right|M_{9}\left|\right|T_{5})$, where $D_{2f}$ is possessed by $U_{i}$ and can be retrieved by the FGWN from $D_{1}$ and its private key $k_{f}$. $U_{i}$ authenticates the FGWN by relying on $A_{f}$ contained in $M_{15}=h(HID_{i}\left|\right|A_{f}\left|\right|D_{1}\left|\right|D_{7}\left|\right|M_{14}\left|\right|T_{8})$, which can be calculated by the FGWN and retrieved by $U_{i}$. $SN_{j}$ verifies the FGWN using $A_{fs}$ contained in $M_{12}=h(SID_{j}\left|\right|r_{fg}\left|\right|A_{fs}\left|\right|D_{1}\left|\right|T_{6})$, which can be only calculated by the FGWN using $k_{f}$ and stored in $SN_{j}$’s memory. The FGWN verifies $SN_{j}$ dependent on $M_{13}=h(SID_{j}\left|\right|r_{fg}\left|\right|A_{fs}\left|\right|D_{8}\left|\right|T_{7})$, where $D_{8}$ is possessed by $SN_{j}$ and can be retrieved by the FGWN from $D_{7}$ and $k_{f}$. $U_{i}$ can verify the legitimacy of SK using $M_{14}$.

#### 4.3.2. Session Key Agreement

$SK=h(D_{1}\left|\right|D_{3}||bD_{1})=h(D_{1}\left|\right|D_{3}||aD_{3})=h(aP||bP||abP)$ is established between $U_{i}$ and $SN_{j}$ in the home region. Similarly, in the foreign region, $U_{i}$ and $SN_{j}$ share a common session key $SK=h(D_{7}||dD_{1})=h(D_{7}||aD_{7})=h(dP||adP)$. The established SK can be used for subsequent communication between $U_{i}$ and $SN_{j}$.

#### 4.3.3. Forward and Backward Secrecy

Forward secrecy is used to guarantee that previously established session keys remain secure in the event that the long-term private keys are compromised. Identically, backward secrecy affords the guarantee that a session key that will be established in the future remains secure even if the long-term private keys are compromised.

The proposed protocol uses the ECDHP to achieve forward and backward secrecy. In the home region, $U_{i}$ and $SN_{j}$ share a common session key SK=h(aP||bP||abP), which is related to the random numbers *a* and *b* generated by $U_{i}$ and $SN_{j}$, respectively. In the foreign region, $U_{i}$ and $SN_{j}$ share a common session key SK=h(dP||adP), which is related to the random numbers *a* and *d* generated by $U_{i}$ and $SN_{j}$, respectively. If all the long-term private keys of $U_{i}$, HGWN, FGWN, and $SN_{j}$ are compromised by an adversary, since the adversary has to resolve the intractable ECDHP to get abP or adP from aP, bP, or aP, dP, respectively, the previous or future session key is still secure. Consequently, forward and backward secrecy can be guaranteed.

#### 4.3.4. User Anonymity and Untraceability

In the proposed protocol, the real identity $ID_{i}$ cannot be acquired by the adversary from the interaction messages. In the home region, there is only the legitimate gateway node who, in possession of private key $k_{h}$, can calculate $D_{2}$ to recover $U_{i}$’s pseudonym $HID_{i}$ and sensor’s identity $SID_{j}$. Simultaneously, considering the one-way nature of the hash function, it is difficult for the adversary to acquire $HID_{i}$ from $M_{3},M_{8}$ and $SID_{j}$ from $M_{5},M_{6}$, respectively. In the foreign region, the adversary without gateway node’s private key cannot compute $D_{2}$ to recover $HID_{i}$. Likewise, considering the one-way nature of hash function, the adversary is unable to get $HID_{i}$ from $M_{3},M_{8},M_{10},M_{15}$. As a result, user anonymity can be achieved. In addition, because of the login request message being updated at each session round, the adversary is unable to trace a specific user. Therefore, the user’s untraceability is guaranteed.

#### 4.3.5. Illegal Login Detection

A user needs to input their identity, password, and biometric information to complete login, and if the terminal declines this session, at least one of these three items is incorrect. In our protocol, when the incoming information is invalid, the identification parameter $B_{i}$ cannot be recovered correctly, which leads to the login request being aborted by the terminal. This mechanism guarantees the system can check illegal login requests quickly.

#### 4.3.6. Stolen Smart Card Attack

The secret parameters $\left\{A_{i},B_{i},C_{i},\theta_{i}\right\}$ are stored in $U_{i}$’s smart card, where $A_{i}=h(HID_{i}$
$\left|\right|k_{h}\left|\right|r_{h})⊕HID_{i}$, $B_{i}=h(HID_{i}||HPW_{i}\left|\right|r_{h})$, $C_{i}=HID_{i}⊕r_{h}$, and $\theta_{i}$ is generated by $Gen(BIO_{i})$. If $U_{i}$’s smart card is lost and obtained by the adversary, then the adversary can get $\left\{A_{i},B_{i},C_{i},\theta_{i}\right\}$, but they are still unable to acquire the correct identity, password, and biometric key data. The adversary cannot compute a correct $HID_{i}$ through $C_{i}$ without $r_{h}$. The biometric key data $\sigma_{i}$ also cannot be recovered correctly without a real $BIO_{i}$. Furthermore, even in this case, there is no chance for an adversary to get the password. As a result, the login request message $M_{1},M_{2},M_{3}$ cannot be figured out without the correct $HID_{i}$. Our protocol can be resistant to stolen smart card attack.

#### 4.3.7. Replay Attack

The timestamp mechanism is used to guarantee the freshness of transmitted messages in our scheme. When the message is exchanged, the node first checks whether the time difference between the received timestamp and its own timestamp is within the acceptable maximum delay allowed by the system. Expired messages will be rejected. As a result, the protocol is capable of defending against replay attack.

#### 4.3.8. Privileged Insider Attack

During the registration phase, user transmits $\left\{HID{}_{i},HPW_{i}\right\}$ to the HGWN via a secure channel. It is assumed that an internal malicious privileged node who executes privileged insider attack in order to get user’s password $PW_{i}$ after getting $\left\{HID{}_{i},HPW_{i}\right\}$. However, the obtained values are hash values consisting of password and biometric key data. Considering the one-way nature of the hash function, it is intractable for the privileged node to extract $PW_{i}$ from $HPW_{i}$. Therefore, our protocol can be resistant to privileged insider attack.

#### 4.3.9. Desynchronization Attack

In the proposed protocol, the user does not store the same secret values with the gateway node. All participants in the protocol are not required to update any information when a session is accomplished. Accordingly, the protocol can resist a desynchronization attack.

#### 4.3.10. Impersonation Attack

In our protocol, in order to forge a user, a valid login request $\left\{M_{1},M_{2},M_{3},D_{1},T_{1}\right\}$ is necessary. Nevertheless, the adversary has no capacity to figure out the true $M_{1},M_{2},M_{3},D_{1}$ without the correct $HID_{i},SID_{j},A_{h},D_{2}$. As a result, the adversary fails to impersonate a legitimate user.

In addition, when the fake gateway node receives the correct login request, it cannot retrieve the true $D_{2}$ without the real private key. Therefore, the adversary is also unable to impersonate a legitimate gateway node.

Moreover, if the adversary wants to forge a sensor node, they need to recover $r_{hg}$ and generate $M_{5},M_{6}$, which all depend on $A_{gs}$ that is only computed by the HGWN and stored in the sensor’s memory. Consequently, this scheme is protected against a sensor impersonation attack.

## 5. Performance and Security Comparison

In order to illustrate the balance between the security and usability of our protocol, the comparative consequences of the security and overhead of our scheme with other associated schemes are as follows, where Case-1 and Case-2 represent the protocol designed in the home region and the foreign region, respectively. According to [17,29,30,31], all operations were implemented in MATLAB on a four-core, 3.2 GHz computer with 8 GB of memory.

### 5.1. Security Features Comparison

The statistics of the security attributes that each scheme can satisfy are summarized in Table 2, where ✓ represents that this literature can satisfy this corresponding security attribute in Table 2, whereas × represents that it cannot achieve. All the indicators listed in Table 2 were achieved by our scheme. Moreover, none of the studies in the literature [13,17,18,19,20] has the capability to achieve forward and backward secrecy. However, the implementation of ECC in our scheme enables ours to accomplish forward and backward secrecy.

### 5.2. Communication Cost Comparison

In order to calculate the communication cost, we assumed that the identity, random number, hash digest, ECC point, and timestamp were 160 bits, 160 bits, 160 bits, 320 bits, and 32 bits, respectively. Additionally, the symmetric encryption/decryption using AES-128 required 128 bits for a 128-bit plaintext block. We evaluated the communication overhead between our protocol and other relevant protocols [13,17,18,19,20] during the login and authentication phases according to the overall quantity of transmitted messages. Table 3 shows the comparison results. Compared with [19], the transmitted number of messages was identical to our scheme, and there were similar communications costs as ours, but our scheme met more security attributes. As we can see, in order to compare with previous protocols [13,17,18,19,20], we chose SHA-1 [32] as the hash function. However, to achieve more security, we recommend using SHA-256 [32] as the hash function.

### 5.3. Computation Cost Comparison

Table 4 lists the approximate required computational time of various cryptographic operations, which were used as a comparative standard. Table 5 compares the computational overhead of our scheme and other relevant schemes during the login, authentication, and key agreement phases. The total cost of the proposed scheme increased slightly. Nevertheless, most of the cost was calculated on the gateway side with strong computational power rather than the resource-limited sensor side. Accordingly, integrated with both security and communication cost, our protocol was relatively secure with an acceptable overhead.

## 6. Conclusions

In this paper, we designed an authentication protocol based on ECC using three factors, applied to the IIoT environment. The proposed scheme was appropriate for single-gateway scenarios, and we also extended it to multigateway scenarios. Furthermore, forward and backward secrecy was realized in our scheme utilizing the intractable ECDHP. The formal security analysis under the ROM indicated that the proposed protocol was able to satisfy semantic security. We simulated our scheme using the formal verification tool Scyther, and the result showed that our scheme was secure. The informal security analysis proved our protocol was capable of satisfying most common security properties. Finally, compared with other representative protocols, the comparative results of security attributes, communication, and computation cost in Table 2, Table 3 and Table 5 clearly showed that our protocols could achieve many security attributes at a reasonable computation cost.

## Figures and Tables

**Figure 1 sensors-22-07510-f001:**
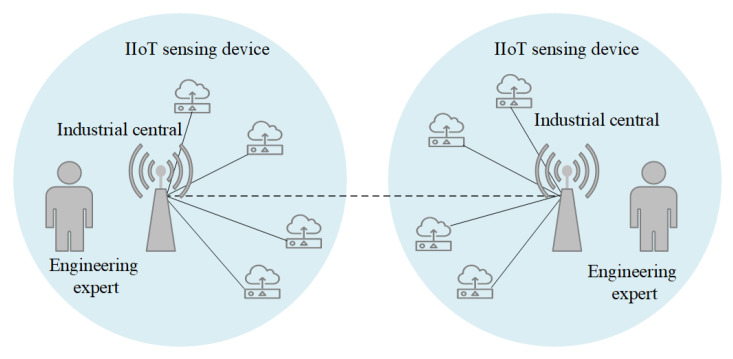
Architecture for an IIoT.

**Figure 2 sensors-22-07510-f002:**
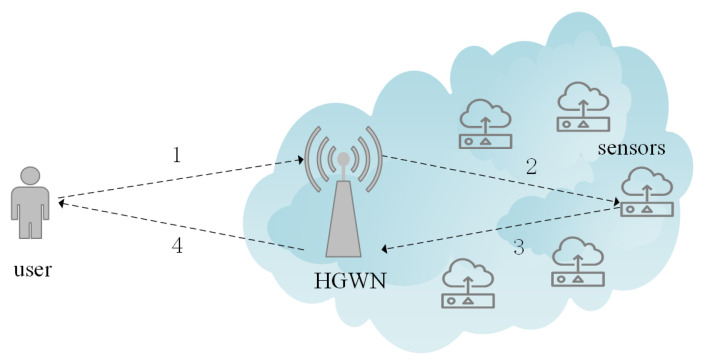
Single-gateway model.

**Figure 3 sensors-22-07510-f003:**
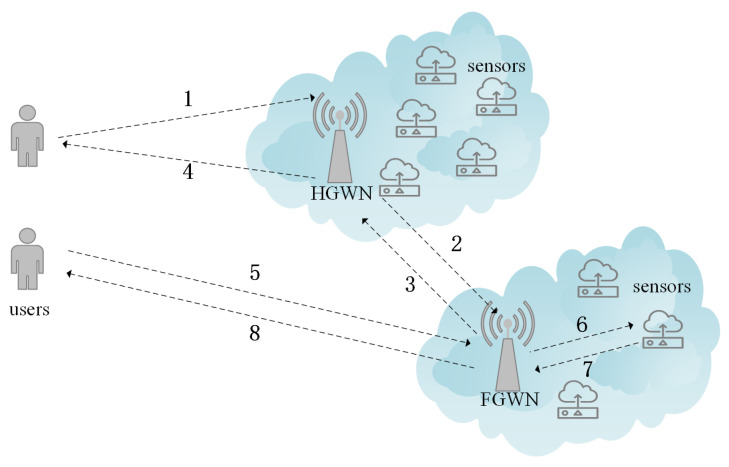
Multigateway model.

**Figure 4 sensors-22-07510-f004:**
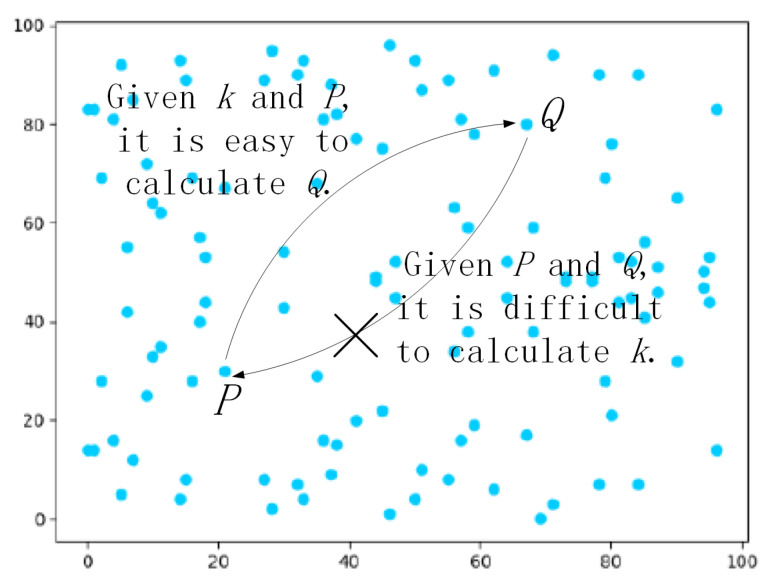
Points over the elliptic curve.

**Figure 5 sensors-22-07510-f005:**
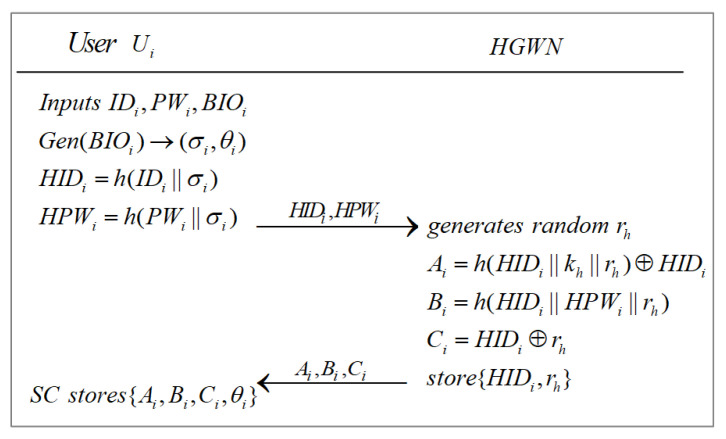
User registration phase.

**Figure 6 sensors-22-07510-f006:**
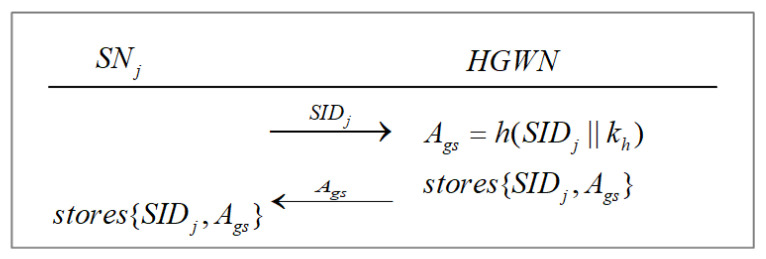
Sensor registration phase.

**Figure 7 sensors-22-07510-f007:**
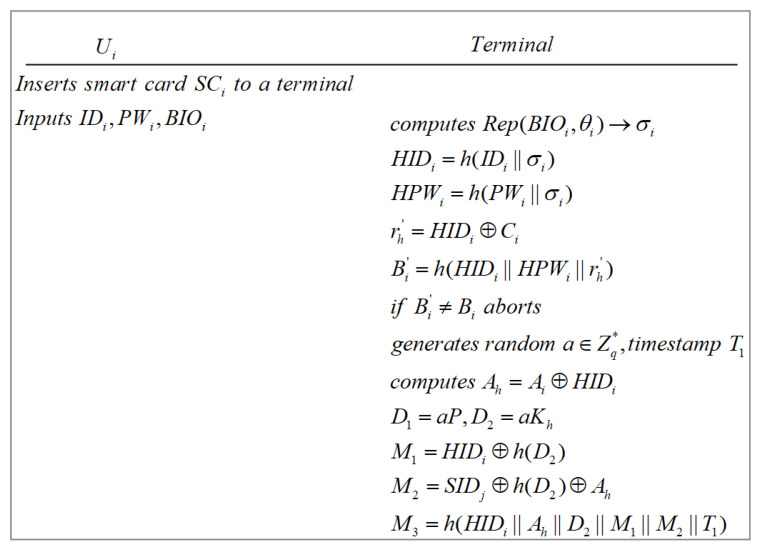
User login phase.

**Figure 8 sensors-22-07510-f008:**
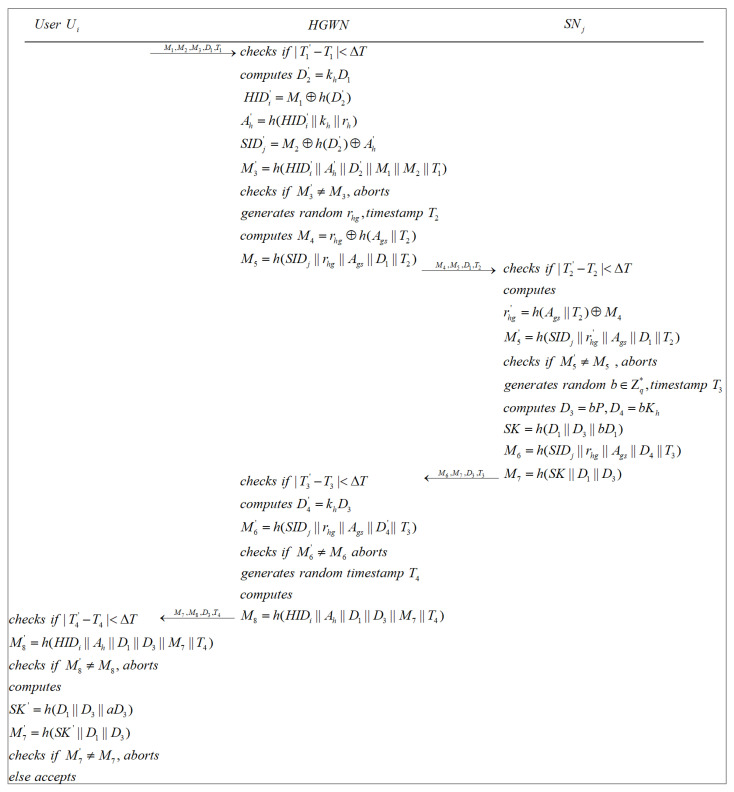
Authentication and key agreement in the HGWN.

**Figure 9 sensors-22-07510-f009:**
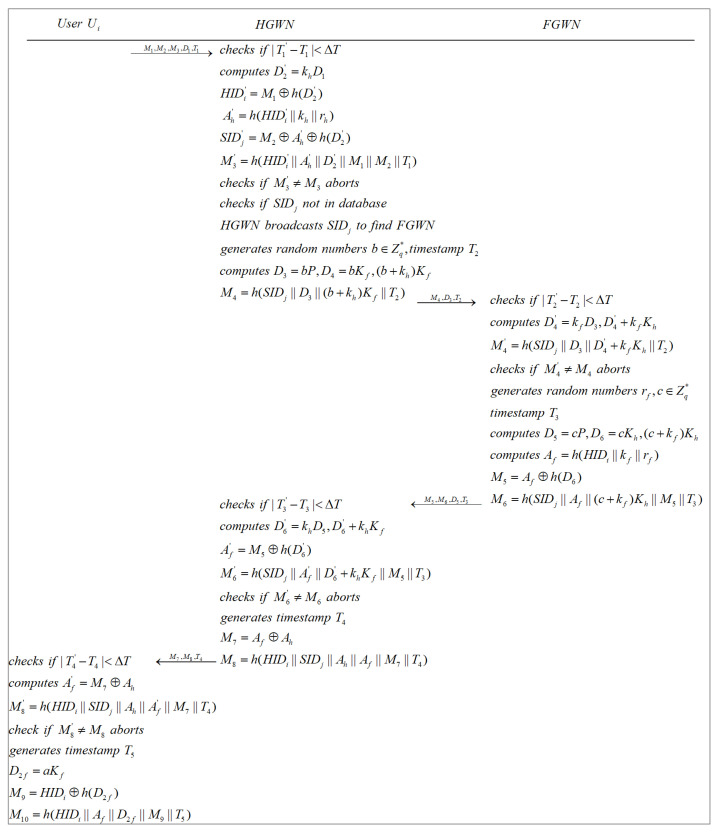
Authentication and key agreement phase 1 in the FGWN.

**Figure 10 sensors-22-07510-f010:**
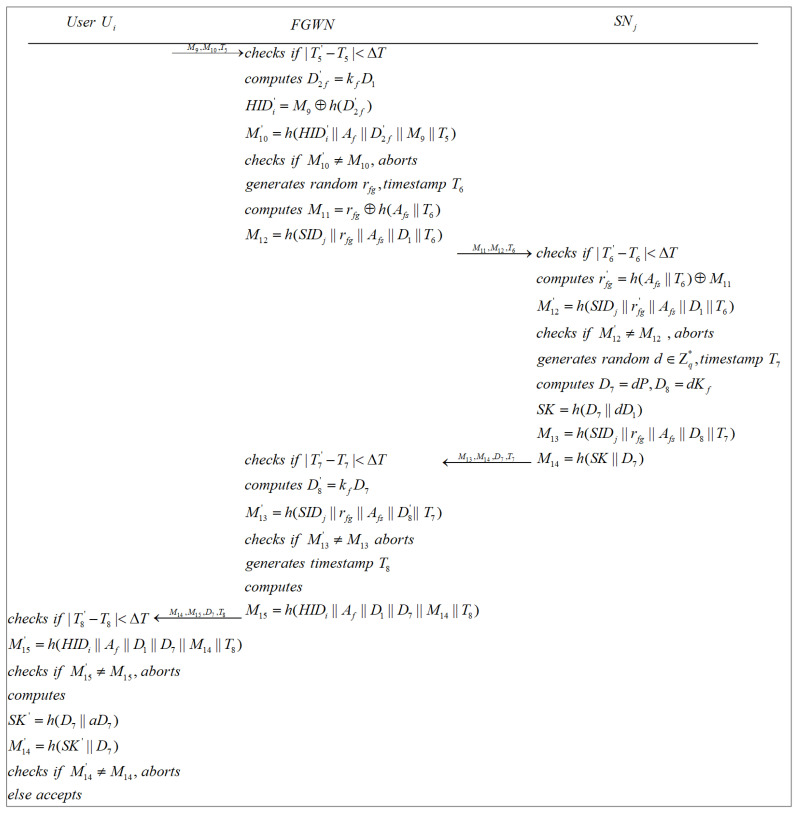
Authentication and key agreement phase 2 in the FGWN.

**Figure 11 sensors-22-07510-f011:**
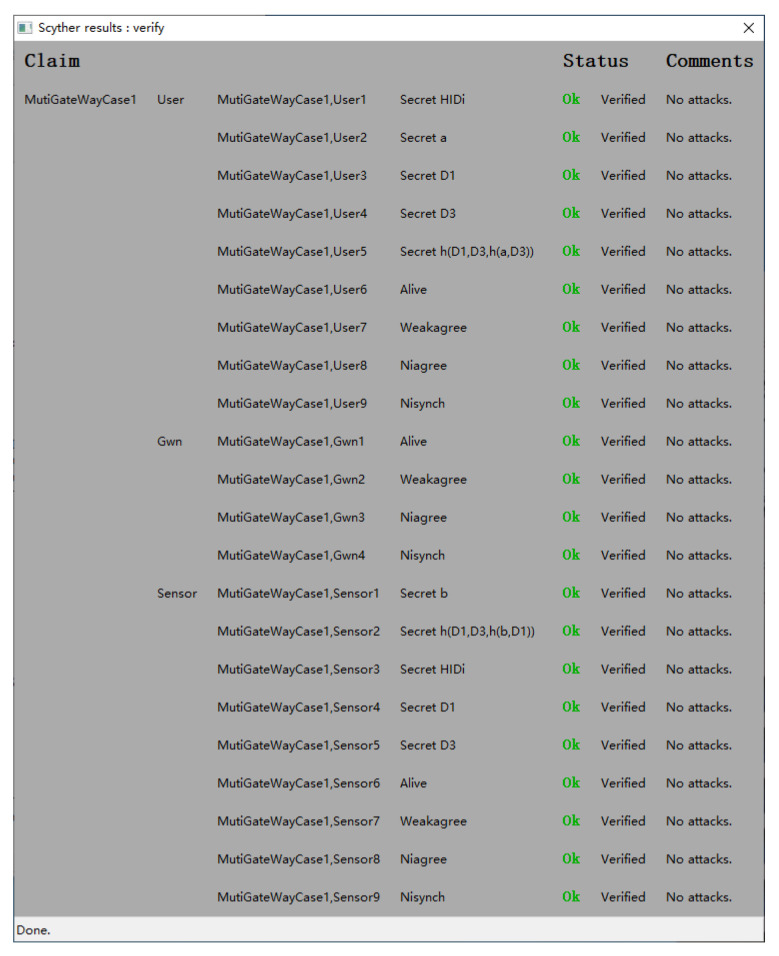
Simulation result in HGWN.

**Figure 12 sensors-22-07510-f012:**
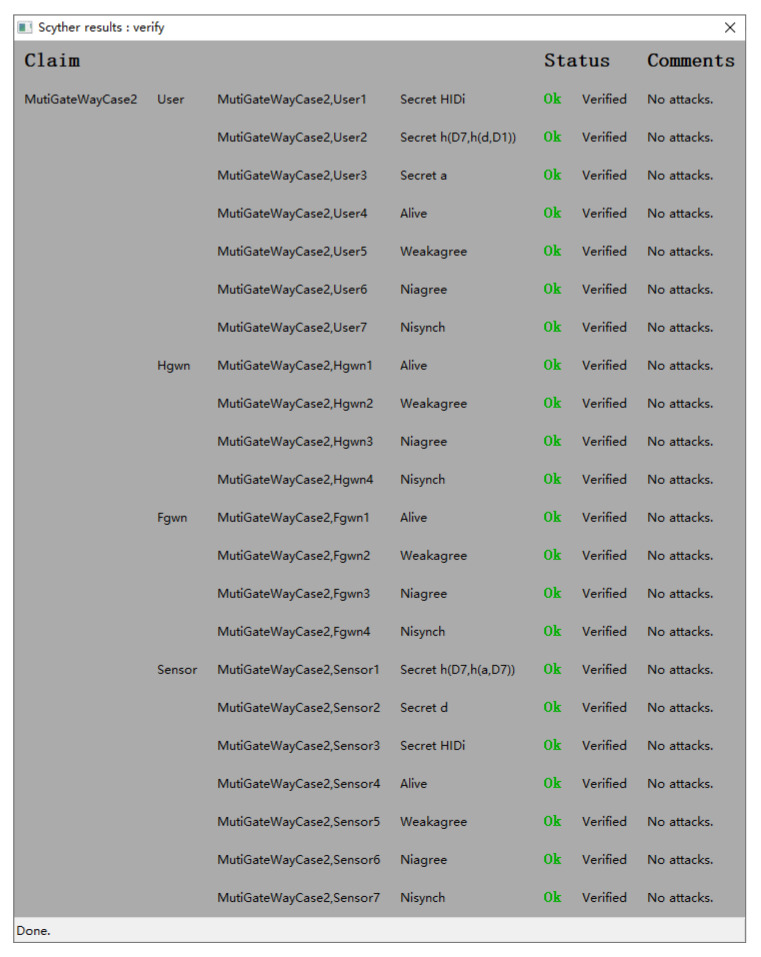
Simulation result in FGWN.

**Table 1 sensors-22-07510-t001:** Symbol description.

Symbol	Description
SA	System administrator
$U_{i}$	*i*th user node
${SN}_{j}$	*j*th sensor node
${SC}_{i}$	Smart card of $U_{i}$
HGWN	Home gateway node
FGWN	Foreign gateway node
${ID}_{i}$	Identity of $U_{i}$
${SID}_{j}$	Identity of ${SN}_{j}$
${PW}_{i}$	Password of $U_{i}$
${BIO}_{i}$	Biometric information of $U_{i}$
$k{}_{h},K_{h}$	Private key and public key of HGWN
$k{}_{f},K_{f}$	Private key and public key of FGWN
$r_{h},r_{hg},r_{f},r_{fg}$	Random numbers
a,b,c,d	Random numbers $\in Z_{q}^{*}$
*P*	A point on the elliptic curve
$T_{1},T_{2},...,T_{8}$	Timestamps
ΔT	Acceptable maximum transmission delay
SK	Session key
h()	One-way hash function
⊕	Exclusive-or operation
||	Concatenation operation
Gen()	Fuzzy extractor probabilistic generation procedure
Rep()	Fuzzy extractor deterministic reproduction procedure

**Table 2 sensors-22-07510-t002:** Security comparison.

Security Properties	[13]	[17]	[18]	[19]	[20]	Ours
Mutual authentication	×	✓	✓	✓	✓	✓
Session key agreement	✓	✓	✓	×	×	✓
Forward and backward secrecy	×	×	×	×	×	✓
User anonymity	✓	×	×	×	×	✓
Untraceability property	×	×	✓	×	×	✓
Illegal login detection	×	✓	✓	×	✓	✓
Stolen smart card attack	×	×	✓	✓	✓	✓
Replay attack	✓	✓	✓	✓	✓	✓
Insider attack	✓	✓	✓	×	✓	✓
Desynchronization attack	×	×	✓	✓	×	✓
Impersonation attack	×	×	✓	×	✓	✓

**Table 3 sensors-22-07510-t003:** Communication cost comparison.

Scheme		Number of Messages	Communication Cost (bits)
[13]	Case-1	2	1504
[17]	Case-1	4	2528
	Case-2	5	3008
[18]	Case-1	3	2784
	Case-2	6	4704
[19]	Case-1	4	2688
	Case-2	8	4480
[20]	Case-1	4	2368
	Case-2	7	3904
Ours	Case-1	4	2848
	Case-2	8	4416

**Table 4 sensors-22-07510-t004:** Execution time of various cryptographic operations.

Symbol	Description	Approximate Computation Time (s)
$T_{h}$	Hash function	0.00032
$T_{ecm}$	ECC point multiplication	0.0171
$T_{eca}$	ECC point addition	0.0044
$T_{sym}$	Symmetric encryption/decryption	0.0056
$T_{fe}$	Fuzzy extractor function	0.0171

**Table 5 sensors-22-07510-t005:** Computational cost comparison.

Protocols		User	HGWN	FGWN	Sensor	Total (s)
[13]	Case-1	$4T_{h}+2T_{ecm}+1T_{eca}$	$4T_{h}+6T_{ecm}+3T_{eca}$	-	$3T_{h}+2T_{ecm}+2T_{eca}$	0.20092
[17]	Case-1	$7T_{h}$	$8T_{h}$	-	$5T_{h}$	0.00640
	Case-2	$8T_{h}$	$1T_{h}$	$7T_{h}$	$5T_{h}$	0.00672
[18]	Case-1	$9T_{h}+1T_{fe}+1T_{sym}$	$5T_{h}+2T_{sym}$	-	$3T_{h}+1T_{sym}$	0.04494
	Case-2	$10T_{h}+1T_{fe}+2T_{sym}$	0	$5T_{h}+2T_{sym}$	$4T_{h}+1T_{sym}$	0.05118
[19]	Case-1	$9T_{h}$	$11T_{h}$	-	$4T_{h}$	0.00768
	Case-2	$11T_{h}$	$7T_{h}$	$7T_{h}$	$4T_{h}$	0.00928
[20]	Case-1	$10T_{h}$	$14T_{h}$	-	$7T_{h}$	0.00992
	Case-2	$14T_{h}$	$6T_{h}$	$17T_{h}$	$6T_{h}$	0.01376
Ours	Case-1	$9T_{h}+1T_{fe}+3T_{ecm}$	$8T_{h}+2T_{ecm}$	-	$5T_{h}+3T_{ecm}$	0.16094
	Case-2	$12T_{h}+1T_{fe}+4T_{ecm}$	$8T_{h}+6T_{ecm}+2T_{eca}$	$10T_{h}+7T_{ecm}+2T_{eca}$	$5T_{h}+3T_{ecm}$	0.38780

## Data Availability

Not applicable.

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
