# Peer review of "Practical Three-Factor Authentication Protocol Based on Elliptic Curve Cryptography for Industrial Internet of Things"

_sensors, 2022, doi:10.3390/s22197510_

Round 1
Reviewer 1 Report
It is needed to check again the English and typos in the text. For example, on line 33, it appears "intrustral central", but on Fig. 1. below appears "industrial central". Pay attention to the acronyms used in the text, for example, "AES" on line 67 is not described.
The mathematical model is hard to check and validate and I suggest inserting some diagrams in the text to make it understandable for readers (besides Figures 2-7 that I do not consider to be named as figures - maybe they can be redesigned as logical schemes).
The Conclusions section can be improved and the authors can describe in detail how their protocols can achieve many security attributes at a reasonable computation cost.
Reviewer 2 Report
-
This paper proposes a protocol for authentication and session key management in IIoT environments. The solution proposed by the Authors is essentially a key management algorithm similar to the likes of Kerberos. The difference between this solution and Kerberos is the use of asymmetric cryptography at HGWN instead of symmetric.
Comments:
- This paper is hard to read. The variables used were not properly defined, and readers will have a hard time understanding what they mean.
- No explanation about how this solution will work in a multi-gateway environment.
- It is not clear how the session keys will be generated after U and SN have been authenticated by HGWN.
- For the communication between HGWN and SN, how long will the session last? What will be the procedure for rekeying?
- What is the benefit of having 3-factor authentication over 2-factor authentication in an environment that emphasizes extremely low latency?
Reviewer 3 Report
In this paper Practical Three-Factor Authentication Protocol Based on Elliptic Curve Cryptography for Industrial Internet of Things is proposed by authors which is very interesting, however for further improvement the authors need to utilised the following points.
1. The authors need to add some more explanations about elliptic curve cryptography.
2. The authors need to add some more explanations into the introduction section because it seems too short.
3. Network model is missing here, it is a need to include a network model with suggested wireless communications technology and IIoT protocols.
4. Threat model is missing here, it is a need to include a threat model with wireless communications technology channel capacity, type of channel, and attacker with his attack capabilities.
5. The authors need to explain that they have choose which type of hash functions and how much bits it would be consumed.
6. The "Performance and Security Comparison" is too short, the authors needs to add the experimental setup like the scheme major operations is tested in which hardware and software resources.
Round 2
Reviewer 2 Report
The manuscript has been sufficiently improved.
Reviewer 3 Report
All my previous comments are utilized very well. i recommend to publish this paper in current for.